# IMPROVED MODELING OF COMPLEX SYSTEMS USING HYBRID PHYSICS/MACHINE LEARNING/STOCHASTIC MODELS

## ABSTRACT

Combining domain knowledge models with neural models has been challenging. End-to-end trained neural models often perform better (lower Mean Square Error) than domain knowledge models or domain/neural combinations, and the combination is inefficient to train. In this paper, we demonstrate that by composing domain models with machine learning models, by using extrapolative testing sets, and invoking decorrelation objective functions, we create models which can predict more complex systems. The models are interpretable, extrapolative, data-efficient, and capture predictable but complex non-stochastic behavior such as unmodeled degrees of freedom and systemic measurement noise. We apply this improved modeling paradigm to several simulated systems and an actual physical system in the context of system identification. Several ways of composing domain models with neural models are examined for time series, boosting, bagging, and auto-encoding on various systems of varying complexity and non-linearity. Although this work is preliminary, we show that the ability to combine models is a very promising direction for neural modeling.

## 1 INTRODUCTION

Modeling has been used for many years to explain, predict, and control the real world. Traditional models include science/math equations, algorithms, simulations, parametric models which capture domain knowledge, and interpolative models such as cubic splines or polynomial least squares among others which do not have explanatory value but can interpolate between known values well. The nonpredictable part of the signal is captured by a stochastic noise model.

The domain/physical models predict $n$ $l$-dimensional output vectors, $\boldsymbol{Y} \in \mathbb{R}^{n \times l}$ given $n$ $k$- dimensional input vectors, $\boldsymbol{X} \in \mathbb{R}^{n \times k}$ with adjustable parameters, $\theta$ used to obtain the best fit (first term in Eq. 1). The unmodeled non-deterministic part of the data is often attributed to random noise fit to various stochastic models $\mathcal{N}(\phi)$ with parameters $\phi$ (2nd term, Eq. (1)). This traditional approach has been very successful. The advantages of a good model include high data efficiency, interpretable, the ability to extrapolate to predict outputs from inputs beyond the range of the training input data, and composable (multiple models can be combined to solve more complex problems).

$$\boldsymbol{Y} = f(\boldsymbol{X};\theta) + \mathcal{N}(\phi) + NN(\boldsymbol{X};\boldsymbol{W}) \tag{1}$$

However, this traditional approach has limitations. Complex systems often have degrees of freedom which are not modeled by the traditional models. These unmodeled degrees of freedom or systematic errors of the measurement are not modeled adequately by the noise model. In addition, the parameters of the physical models, $\theta$ can be in error or be time dependent. In these cases the behavior of unmodeled part of the system is not random and thus the usual combined deterministic-stochastic model is inadequate.

Neural models $NN(\boldsymbol{X};\boldsymbol{W})$, e.g. neural network models, are fundamentally just another form of parametric models where $\boldsymbol{X}$ are the inputs and $\boldsymbol{W}$ are the weight parameters. However, neural modes have unique properties to exploit. First, neural models can handle high dimensional input-output relations with complex patterns. Second, like interpolative models such as cubic splines or polynomials, NNs are sufficiently expressive to fit many possible relations but are often not good at

extrapolation, (see below).Trask et al. (2018) Third, neural models do not require handcrafting basis functions. Hence, neural models have the potential for describing unmodeled degrees of freedom, systematic errors, and nonstationary behavior.

In this paper, neural modeling is combined with traditional modeling to achieve the advantages of both traditional and neural models and compensate for the problems mentioned above using following steps. (1) Composing Hybrid Models. We examine several ways of creating hybrid models: boosting, ensemble, and cyclical autoencoder (Fig. 1). Combining domain models and neural models requires assumptions about relationship between various system and noise. For example, Eq. (1) makes an implicit assumption that the models are composed by addition but there are many other possible assumptions. Unlike most boosting approaches, we use different model classes and loss functions for the various stages. (2) Extrapolation Testing. An extension to the traditional machine learning approach of dividing the data set into test and training portions is extended to include both interpolative and extrapolative testing sets as a stringent test of modeling power. (3) Stochastic Loss. Unlike previous approaches, the quality of the hybrid models to produce truly stochastic residuals is enforced using novel loss functions that enforce appropriate correlation of residuals.

In this work, this paradigm is applied to system-identification (SysID) for simulated and real systems. The results demonstrate that these models decompose into deterministic, predictable, and stochastic components and can handle more complex systems

## 2 RELATED AND PREVIOUS WORK

The use of a combination of traditional models and neural models has been investigated before. Psichogios & Ungar (1992) used an extended Kalman filtering and a multilayer perceptron. Talmoudi et al. (2008) used Khonen neural networks as form of cluster analysis to help in traditional models. Zhang (2003) used hybrid ARIMA neural network to explain data. Pathak et al. (2018) investigated a liquid state plus physics model hybrid. Most of these papers do not use the latest machine learning approaches in particular, boosting as a general principle for a modeling paradigm or the most effective current neural models. In this earlier work, the models are evaluated by MSE over an interpolative testing set (discussed below). Thus in previous work, the flexibility of neural models to interpolate with enough parameters and training has resulted in the conclusion that end-to-end neural models without domain/physics modeling nearly always exhibit the lower MSE and therefore there is no advantage for incorporating domain knowledge. In addition, the important issues of how to compose the hybrid models, evaluate their ability to incorporate prior knowledge, to provide interpretability in terms of modeled, unmodeled predictable, and unpredictable components, and to evaluate their ability to extrapolate vs interpolation, were not adequately addressed. These relatively new issues are addressed in this work.

Various approaches for composing models include following. Jacovi et al. (2019) generated supervised data using a blackbox physics model to fit a neural model used to back propagate the errors to the input models. Innes et al. (2019),Innes (2018) propagate the errors through parametric physics models using automatic differentiation. This works for small models but doesn't scale well, solve the more complex domain problems, address extrapolation or exhibit noise model consistency. Sahoo et al. (2018) introduces the idea of specific functional neurons computing model function which slow computation by mixing NN and nonlinear least squares as well as limited modeling capability. Amos & Kolter (2017) constructed a nonlinear least squares layer to fit several interactions of a model as part of a larger neural network. Other work Zhou et al. (2017) incorporated a physics model by adapting the inputs to the model using a neural network to adjust the outputs to be the desired results. None of these address the issue of training protocols, interpreting the neural component, or investigating the statistical properties of the residuals. Bengio et al. (2006) looked at boosting one stage of a neural layer and Finally, Huang et al. (2017) has investigated the problem of composing through boosting. Like most boosting, the model class and loss function where the same for each stage and no attempt to connect with the larger modeling paradigm was attempted.

## 3 COMPOSING HYBRID MODELS

To form hybrid models, we consider a few promising ways combine a domain model with neural networks and stochastic models: a) sequential, b) parallel, and c) cyclic training as shown in Fig. 1.

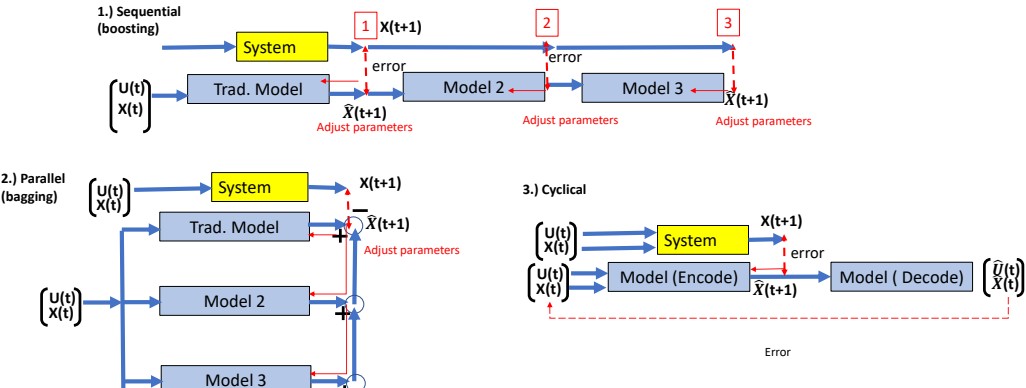

Figure 1: Various ways of creating hybrid domain/neural models. 1) Sequential boosting. 2) Parallel ( bagging or ensemble) and 3) Cyclic (variational autoencoding training

## 3.1 SEQUENTIAL METHOD

The sequential or boosting method generalizes the traditional modeling splitting the output into deterministic and stochastic components. In this model composition procedure, the parameters, $\theta$ of one or more domain/physics models are trained by selecting a loss function discussed in the next section. The next stage is implemented by selecting another model using residuals and and previous inputs as the new input and selecting a loss to boost (train) the next stage parameters. This process can be iterated a number of times. Normally boosting uses one model class such as trees but importantly in our case, the weak learner is the domain model(s) for the first stages followed neural model boosting to create a hybrid. The final residuals are ascribed to stochastic models. More precisely, the boosting method is implemented as follows. A sequence of $M$ learners is used to refine the predicted output. The weak or base models consist of physics/deterministic models which are followed by NN models to capture the predictable but complex portion

$$\widehat{f(x)} = \hat{f}^M(x) = \sum_{i=0}^{M} \hat{f}_i(x) \tag{2}$$

where $M$ is the number of iterations, $\hat{f}_0$ is the initial guess and $\hat{f}_i$ are the boosts.

$$\hat{f}_t(x) \longleftarrow \hat{f}_{t-1}(x) + h_t(x, \theta_t) \tag{3}$$

where

$$\theta_t = arg\ \min_{\theta}\left[\Psi_t\left(y, \hat{f}_{t-1}(x) + h_t(x, \theta)\right)\right] \tag{4}$$

where $\Psi_t(.,.)$ is a loss function for stage $t$ discussed below. In summary, the sequential method involves a number of stages where a model is fitted to the residuals using a stage specific loss function. Generally, the domain models are applied first followed by the neural models.

## 3.2 ENSEMBLE METHOD

In the parallel (ensemble) modeling (Fig. 1(2), the various models are trained in one stage to minimize the loss between the system and the model. Both $\theta$ and $W$ are trained simultaneously. This has the advantage of finding a global optimum but is more compute intensive.

## 3.3 CYCLIC METHOD

In this hybrid method, for each training stage, a model is selected with a loss function as in sequential, but a second decoder NN model is learned as well which provides a cyclical loss term to replicate the input. This cyclical training is performed for each stage. So the method is similar to the sequential method in terms of stages of domain then NN learning. This method often improves training and provides a decoder/inverse model of use inverse modeling and is related to the implicit deep equilibrium layers of Bai et al. (2019). There are of course other ways compose models but these methods provide a direct connection with the additive decomposition of Eq. (1).

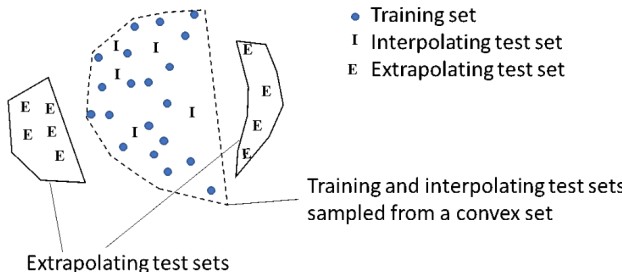

Figure 2: Interpolating training and test sets, and extrapolating test sets defined by a complex hull

## 4 STOCHASTIC MODELING

In order to fit or the parameters, the loss function, $\Psi(.,.)$ for each learning/fitting stage must be selected. For the domain model training, the ensemble model or the first stages, the usual MSE function

$$\Psi_t\left(y, f\left(x\right)\right) = \sum_{i=1}^{N} r_i^2 = \rho_0 \tag{5}$$

is sufficient where $\rho_i$ is the $i$th lag of the autocorrelation function of the residual

$$r_i = y_i - f(x_i) \tag{6}$$

In general however, the MSE does not insure that the residuals after training are compatible with the stochastic model. The stochastic model normally assumes independent, identical distributed (IID) random variables. The residuals are compared to the stochastic model using an autocorrelation function usually a zero mean Gaussian with variance $\sigma^2$. Introducing a new objective namely a Ljung-Box (LJB) loss function

$$\Psi_t\left(y, f\left(x\right)\right) = n(n+2) \sum_{k=1}^{L} \frac{\rho_k^2}{n-k} \tag{7}$$

ensures uncorrelated residuals and compatibility with the stochastic model. $L$ is a hyper parameter which should be larger than possible correlations. If residual correlations are expected, the expected correlation can be built into the LJB function. Thus, to create and train a hybrid model, domain and NN models are selected. For each stage of learning, a loss function is selected to drive the outputs towards a desired goal. Decorrelation loss functions are used to control the correlations of the residuals.

## 5 INTERPOLATIVE TRAINING/EXTRAPOLATIVE TESTING

Finally, having selected models and a loss function, the training and testing data must be selected. For the most part, previous work has sampled the testing set from the same input region as the training set. Because of the interpolating ability of large parameter neural models, the models achieve a lower MSE than domain and hybrid domain/neural models. But this measure of quality does not measure the extrapolation ability of domain models. Thus, a new method for creating testing data is used. (Fig. 2). An interpolating test set is created by sampling from the convex hull at different points than the training set. An extrapolating test set is created by sampling outside the training convex hull. Extrapolative testing identifies those models which generalize to larger input domains and discriminates against those models which memorize the data. Indeed, in the results below, high parameter models look good for interpolative but not for extrapolative data.

The alternative of performing end-to-end of all the models at once is an significant alternative. However, this procedure is often very inefficient because the switch from neural models to nonlinear

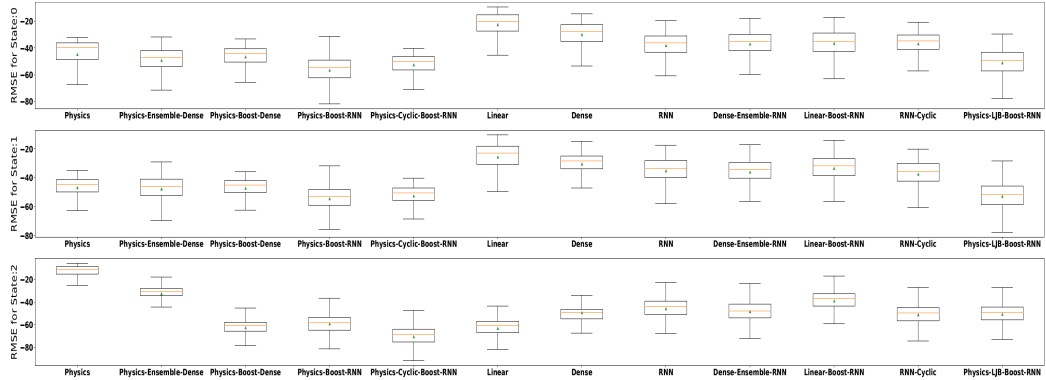

Figure 3: RMSE in db for the inverse pendulum problem using an interpolative test set for various models, model hybrids, and loss functions during training as described in the text

domain models is not an efficient use of GPUs. Moreover, one can guide the output of each stage and maintain intepretability by training in stages.

In summary, the modeling procedure selects one or more models, combines them in parallel or sequentially, selects a training set, and uses a loss function to train this stage. The process is repeated, preferable with complementary models to the previous ones and another loss function, fit to the residuals or outputs of the previous stage. A decorrelation loss function insures consistent Eq. (1) decomposition.

## 6 RESULTS

The above process for modeling complex phenomena is applied to system identification for a number of different mechanical problems in time series prediction, an area where machine learning has not had as much success Makridakis et al. (2018).

In particular, problems from the OpenAI gym (Brockman et al. (2016)), the inverse pendulum and a double pendulum are simulated as a function of time with Gaussian noise added. Finally, the procedures are applied to an actual DC motor system with backlash. In order to create a system-id data set, a series of actuations excite the system, and the resulting response simulated or measured. Training and testing data sets consist of the previous state $x(t+1)$, the actuation $u(t)$, and residuals $r(t)$. The outputs are the next time step residual values $r(t+1)$. The state of each system includes various degrees of freedom such as the angle, angular velocity, and acceleration. The actuation time series include a training set and an interpolative testing set from a restricted range of actuations restricting amplitude and/or frequency. The extrapolative test set is created by increasing the amplitude and/or the frequency of actuation for the various models and the actual physical motor. The training time series is used to fit/train various models including a physics model, a linear state space model, a dense neural network, an recursive neural network (RNN) and a cyclical model. The physics model and the linear state space model are also used as the first stage in hybrid models and RNN, dense NN and cyclical NN are used as to fit the residuals with either MSE or LjungBox error (LJB) as the objective function.

### 6.1 INVERSE PENDULUM

The inverted pendulum swingup problem is a classic problem in the control literature. To create a physics model we introduce some errors into the model (mass of the pendulum was changed to 5 from 1 and length change to 10 from 1) to insure an error to be captured by other models. Otherwise, the physics model fits nearly perfectly. Model inputs are: $\cos\Theta(t), \sin\Theta(t), \dot\Theta(t), U(t)$ (Applied Torque) and outputs are $\cos\Theta(t+1), \sin\Theta(t+1), \dot\Theta(t+1)$

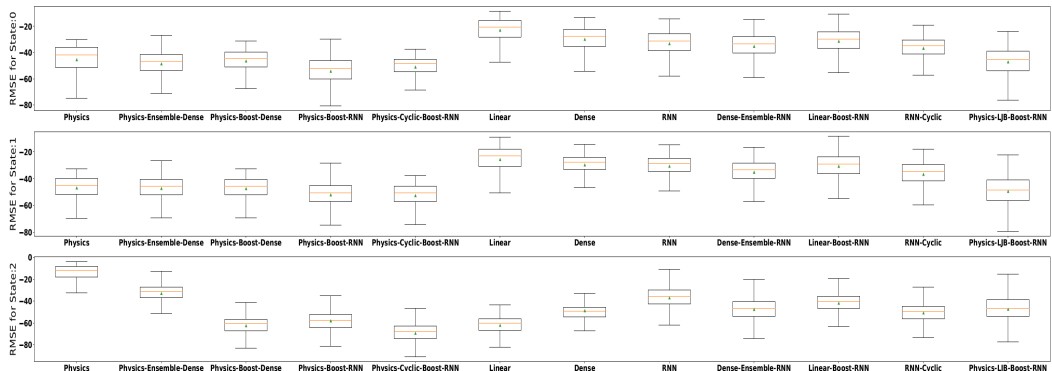

Figure 4: RMSE in db for the inverse pendulum problem using an extrapolative test set for various models, model hybrids, and loss functions during training as described in the text

We train our models on a constrained training dataset of size 25000 points with control actions between -1.0 to 1.0. The performance of different model compositions on a interpolative validation dataset is shown in Fig. 3.

We then create an extrapolation dataset of size 25000 utilizing the entire space of control action space i.e. between -2 to 2. The performances of the models on this dataset is shown in Fig. 4.

In Fig. 3, the box plot of root mean square residuals (RMSE) between the predicted state and the observed state in $db = 20log_{10}|RSME|$ is plotted for various models showing the median, 25/75 percentile and the 5/95 percentile along with the arithmetic mean (triangle) over the testing time period. As expected, the dense NN and the RNN do very well compared with the nonlinear physics model. Boosting the physics model with a linear state space model helps reduce the median but the variation (error bars) is still large. Boosting with a dense NN and in particular, an RNN greatly reduces the median and variance of the residuals. The cyclical autocorrelation also does a good job reducing the interpolative testing errors. The LJB loss boosting results in a worse MSE as expected because the LJB attempts to minized the residual correlations.

For the extrapolative testing set, the models exhibit greater variation(Fig. 4). Except for State 2, the physics models do better than the linear model, and the various single neural network models, Dense and RNN. These results validate the claim that physics models even with incorrect values extrapolate better than NN models. Because db is log, small plotted changes are actually quite significant. In (Fig. 4), even when the NNs or linear are boosted, the median errors are larger. Because the problem is nonlinear, the extrapolated linear model prediction errors are rather large. Only the physics model boosted with an RNN and trained with a LBJ loss function competes with the other physics models. Thus, the combination of the physics plus the neural network models improve on the pure NN models and on the physics models as well. We also note that in general, the MSE for the extrapolative data sets are larger than the interpolative data sets as expected. In summary, the results for the inverted pendulem confirm that hybrid domain(physics) models augemented with NN models results in better fits especially for the extrpolative data sets.

## 6.2 DOUBLE PENDULUM

Next we look at the double pendulem to further demonstrate that the hybrid modeling can capture unmodeled degrees of freedom. We create a simulation of a double pendulum in free fall under gravity. The model system consists of a large pendulum with a small pendulum at the end starting at 90 degrees and oscillates without actuation. The data is interpolative because the testing is within the range of training data. Model inputs are: $\Theta(t), \dot{\Theta}(t), \ddot{\Theta}(t)$ and outputs are $\Theta(t + 1), \dot{\Theta}(t + 1), \ddot{\Theta}(t + 1)$. The physics, dense and RNN fit the state of pendulum as if it were a single large pendulum. The RMSE for the physics model is significantly reduced by boosting the physics model with neural models, in particular the RNN reduces the error (see Fig. 9 in appendix). Fig. 5 shows how the hybrid improves the autocorrelation function of the residuals. In the first set, the autocorrelation function for only the physics model for a single pendulum fit to the double pendulum residuals are plotted.

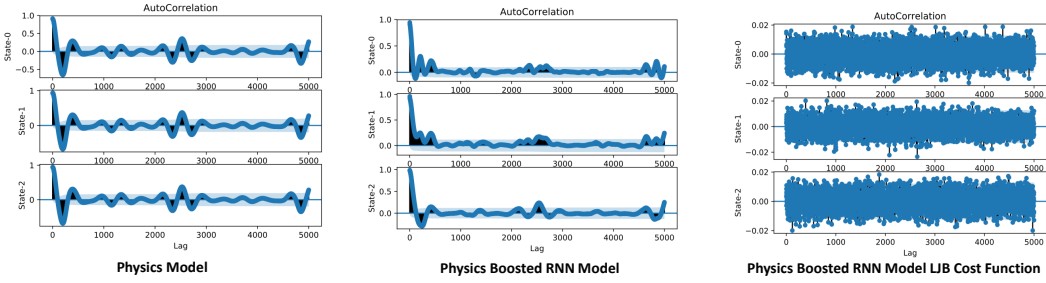

Figure 5: AutoCorrelation the double pendulum physics model–LSE loss function (left), physics model boosted by RNN (middle) and physics model boosted by a RNN trained with Ljung-Box LJB) loss function.

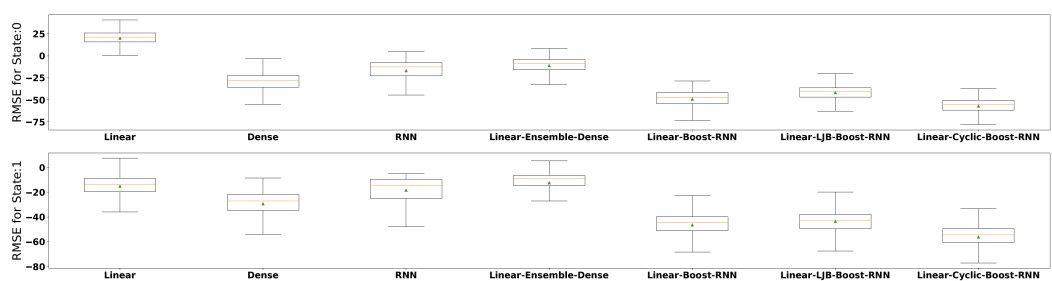

Figure 6: RMSE in db for a real DC motor with backlash for an interpolative test set for various models

Because the second pendulum motion is not modeled, the residual autocorrelation function exhibits the oscillation of the unmodeled motion. In the second panel, the physics model boosted by the RNN with MSE as the loss function shows that the RNN is successfully modeling the oscillation of the second smaller pendulum as the autocorrelation function exhibits reduced amplitude oscillations. When the LJB loss function is used for the RNN boosting phase, the autocorrelation function (with zero lag suppressed) is very close to uncorrelated and the variance of the residuals is in fact the noise added to the simulation. Thus, we have successfully captured the large motion with a physics model, the second pendulum (the unmodeled degrees of freedom) with the RNN, and the unpredictable signal components with a consistent IID stochastic model. This result confirms the ability to capture structured disturbances.

## 6.3 DC MOTOR

Finally, we apply the hybrid modeling to actual physical systems. A DC motor was coupled to a rotary encoder with a 3D-printed shaft. The shaft was divided into two pieces and these two pieces

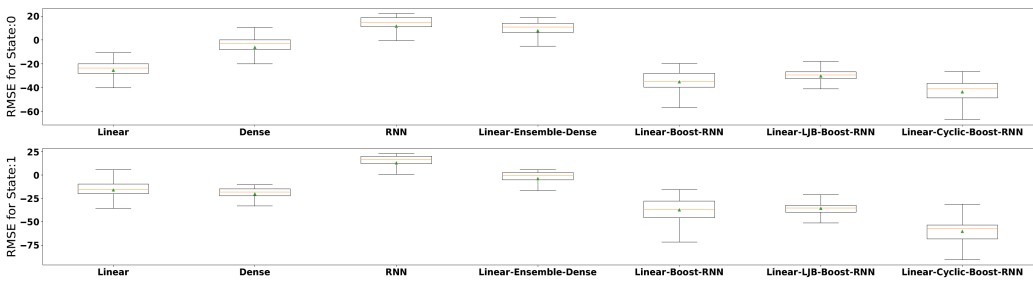

Figure 7: RMSE in db for a real DC motor with backlash for an exterpolative test set for various models

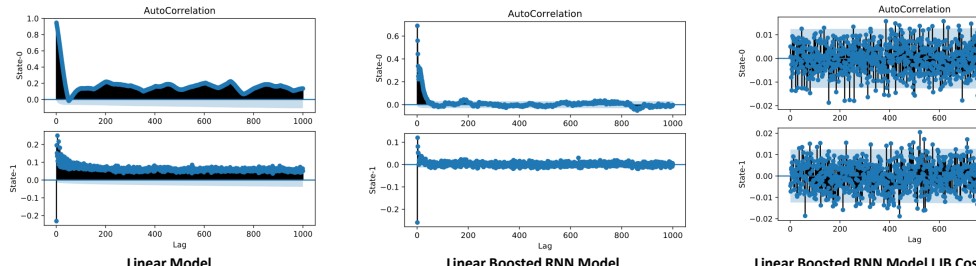

Figure 8: AutoCorrelation residuals for a real DC motor with backlash–a linear(physics) state space model trained with a LSE loss (left), a linear(physics) state space model boosted by an RNN with LSE loss (middle) and a linear(physics) state space model boosted by an RNN with a Ljung Box loss

were coupled through gearing that purposely exhibited substantial backlash. The voltage was controlled through Pulse Width Modulation (PWM) with polarity which allowed for a range of speeds and direction changes. Position and velocity data of the shaft was collected from the encoder using a National Instruments DAQ board. For data collection, the Arduino controller was programmed to randomly vary the speed and polarity of the motor collecting data inputs: $\Theta(t), \dot{\Theta}(t), U(t)$ (Voltage.) paired with model outputs: Next time step $\Theta(t+1), \dot{\Theta}(t+1)$. This input,output pair data was fit to a linear state space model, (i.e. a physics model) where the linear coefficients are interpretable in terms of motor/load moment of inertia, friction, winding resistance, and motor inductance. The data was also modeled using RNN, dense NN, and various hybrid models. Fig. 6 shows that the dense NN and RNN are able to capture some of the backlash. The linear/physics model boosted by an RNN using cyclic training does the best for the interpolative data set with 10 Hz excitation. For the extrapolative data in Fig. 7, the frequency of excitation pulses were 100 Hz. As expected, the RMSE is larger for the extrapolated data set. Boosting by a NN, particularly the RNN, captures the delay of the backlash and the MSE is much smaller. In Fig. 8, the modeling shows how the domain model picks up some of the behavior(left), the RNN captures the backlash (middle), and the LJB loss function forces the residual to match the IID assumptions (right). Thus, these results demonstrate that for both simulated and real systems, the boosting of domain models with neural models utilizing whitening loss functions, results in consistent, interpretable, extrapolating models.

## 7 CONCLUSIONS AND FUTURE DIRECTIONS

Combining neural models with physics(domain) and stochastic models greatly expand the ability to model complex phenomena particularly for control. The expanded hybrid models incorporate the domain knowledge, interpretability, data efficiency and extrapolability of domain models with neural models which can model complex, high dimension, but predictable uncontrolled, unmodeled degrees of freedom of the system and of the measurement system (systematic noise). Using boosting, novel whitening objective functions, and extrapolative/interpolative testing sets, these hybrid models capture the behavior of more complex models in a meaningful decomposition. These models help solve the problems of unmodeled non-stochastic components of system behavior. For future work, measures such as signal-to-noise, error rates etc can be generalized to the case of non-stochastic neural modeled behavior. The combined modeling solves problems with structure noise. In the field of control, stability bounds for control systems can be implemented by using these models in the context of robust control.

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

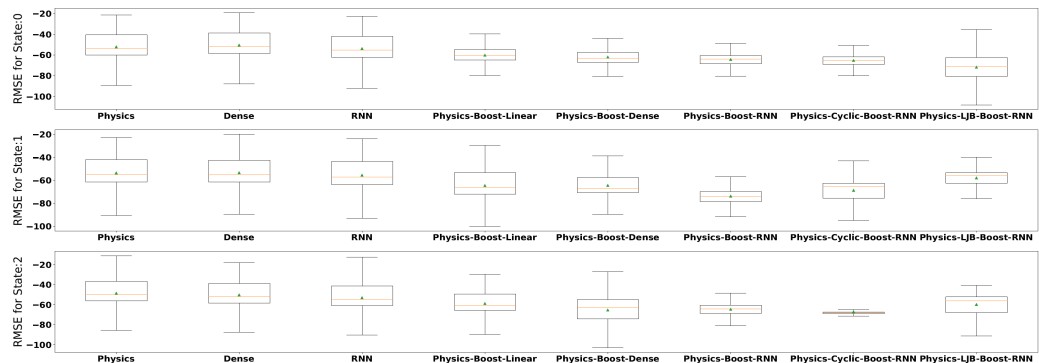

Figure 9: RMSE in db for the double pendulum problem using the interpolative test set for various models, model hybrids, and loss functions during training as described in the text. The problem is treated as a single pendulum with the second pendulum as unmodeled degree of freedom

G Peter Zhang. Time series forecasting using a hybrid arima and neural network model. *Neurocomputing*, 50:159–175, 2003.

Siqi Zhou, Mohamed K Helwa, and Angela P Schoellig. Design of deep neural networks as add-on blocks for improving impromptu trajectory tracking. *56th IEEE Conference on Decision and Control*, arXiv preprint arXiv:1705.10932, 2017.

# Appendices

## A  NEURAL NET MODEL DETAILS

Hyperparameter discussion are presented here. All models are trained on Keras (Chollet et al. (2015)) with TensorFlow (Abadi et al. (2016)) as the backend. Trained models are saved based on performance on a validation dataset created from the Interpolative convex set. The initial learning rate is set at 0.01 and the learning rate is halved if the validation loss plateaus for 10 consecutive epochs. Early Stopping is effected if the models do not show any reduction in validation loss for 30 consecutive epochs. Optimizer used is Adam (Kingma & Ba (2014)) and models are trained with mini batch of size 512. Batch Normalization (Ioffe & Szegedy (2015)) is used at the input stage of all models. All trained Dense Models have 10 Hidden Layers with 256 weights in each layer. All trained RNN models have 10 Hidden layers with 50 recurrent weight blocks in each layer.

## B  DOUBLE PENDULUM RMSE

The interpolative double pendulum results are shown in Fig. 9.

