# OpenReview forum: "Improved Modeling of Complex Systems Using Hybrid Physics/Machine Learning/Stochastic Models"
_ICLR.cc/2020/Conference — Reject_

### Official Review · AnonReviewer3 · 2019-10-21
**Official Blind Review #3**

**Rating:** 1

**Review:**

This paper conducts several experiments to compare the extrapolative predictions of various hybrid models (sequential, ensemble, and cyclic), which compose physical models, neural networks and stochastic models.

Unmodeled dynamics is a bottleneck for model learning, model-based reinforcement learning and sim-to-real transfer. This paper tries to tackle this important research challenge. However, this research is at its early-stage and the results are preliminary. I would not recommend accepting the paper at this time, and encourage the authors to continue this promising research direction. First, the conclusion of the paper is not clear to me. Which is the best way to compose models? Is it physics+RNN+cyclic LJB loss? It would be helpful to put all the visualizations of the results together and organize it in a way that can easily reveal conclusions across different experiments. If the conclusions are inconsistent across different experiments, detailed analysis and explanations are expected. Second, it is not sufficient to just analyze the regression errors of different hybrid models for an ICLR paper. The paper would be much stronger if the it could demonstrate that with the more accurate hybrid model, model-based learning performs better or transfer learning is more straightforward. For example, it could change Section 6.2 to control a double inverted pendulum using only a rough physics model of a single pendulum. Similarly, it could augment Section 6.3 to control a real inverted pendulum with a DC motor with large backlash.

In addition, I have a few questions/comments about the clarity of the writing:
1) Are \theta in eq. (1) the parameters of the physical system, such as mass, length of the rod for the inverted pendulum case?
2) I am not familiar with LJB loss function. If it is commonly-used, please add a reference. If not, more explanations would be helpful. Is LJB loss function used alone or combined with the MSE loss?
3) Is there any particular reason that cos\theta and sin\theta are chosen as states for Inverted Pendulum instead of \theta? In Double Pendulum, \theta is used.
4) In DC Motor, "This input, output pair data was fit to a linear state space model, (i.e. a physics model)..." Is the output linear to the states (\theta, \dot{\theta}) or linear to the physical parameters (inertia, friction, etc.)? In previous examples, such as Figure 4, physics model and linear model are separate. It is confusing to me that in this example, the linear model seems to be the physics model. Is the dynamics equation of the DC motor linear? Am I missing something?

**Experience Assessment:**

I have published one or two papers in this area.

**Review Assessment: Checking Correctness Of Derivations And Theory:**

I assessed the sensibility of the derivations and theory.

**Review Assessment: Checking Correctness Of Experiments:**

I assessed the sensibility of the experiments.

**Review Assessment: Thoroughness In Paper Reading:**

I read the paper at least twice and used my best judgement in assessing the paper.

---

### Official Review · AnonReviewer1 · 2019-10-24
**Official Blind Review #1**

**Rating:** 1

**Review:**

The paper presents approaches for combining neural network (NN) with non-NN models to predict behavior of complex physical systems.
I found this paper very hard to read, with a lot of details and key pieces of information missing or vaguely stated.  Following are some specific instances in no particular order:
a) From the results it seems that often approaches using ‘cyclic-boost’ perform the best.  However the description of cyclic boost in Section 3.3 lacks any precise description of what the cyclic approach does.  For instance, the NN that predicts the inputs from model outputs, how is that used and under what objective is that learnt?
b) The main claim of the paper is that combining physics models with NNs is better than NNs alone, esp. when tested on ‘extrapolative data’.  However, if cyclic-boost is the best approach, the paper should include results with cyclic-boost-RNN combination, these do not seem to be there to assess value add of the physics models in the best performing system.
c) Section 3.2 states a parallel ensemble of NN and non-NN models has the advantage of finding ‘global optimum’ … this is a strong statement and needs to be demonstrated.
d) The description of new loss function in Section 4 is unclear.  For instance, what is ’n’ in (7)?
e) In Section 3.1 I’m assuming that ‘h_t’ denotes the model being trained in iteration ’t’, and some of these are NN whereas others are domain models?

**Experience Assessment:**

I have published in this field for several years.

**Review Assessment: Checking Correctness Of Derivations And Theory:**

I carefully checked the derivations and theory.

**Review Assessment: Checking Correctness Of Experiments:**

I assessed the sensibility of the experiments.

**Review Assessment: Thoroughness In Paper Reading:**

I read the paper at least twice and used my best judgement in assessing the paper.

---

### Official Review · AnonReviewer2 · 2019-10-24
**Official Blind Review #2**

**Rating:** 1

**Review:**

The authors present a new hybrid models that incorporates traditional models with neural networks by boosting and reducing the residuals in stages. They show 3 different boosting schemes: sequential boosting, parallel boosting and cyclical boosting (inspired by variational autoencoders). Furthermore the authors present the new Ljung-Box (LJB) loss function which reduces autocorrelation. They test on 2 simulated systems and on a physical DC Motor and put an emphasis on testing their result on extrapolated (unseen) data.

Strengths:
- The authors combine traditional models with deep learning approaches.
- The models can extrapolate to unseen data better than conventional deep learning approaches.
- The results show that the authors found a loss function that can reduce autocorrelation.

Weaknesses:
- The paper lacks an exact explanation for every model that was used. This is made worse by a lack of consistency, e.g. the model Physics-Boost-Dense doesn't explain which of the 3 boosting schemes is used.
- The authors propose a new Ljung-Box (LJB)  loss function. It contains hyper parameter L "which should be larger than possible correlations". It is not mentioned how to find such a value or what the value for the experiments for. They show that this loss function can reduce autocorrelation, but it comes at the expense of higher RMSE. It is not clear in what scenario this is useful.
- The methodology is difficult to follow. E.g. there is a mathematical explanation in 3.1, but not for 3.2 or 3.3. In Fig. 1(1) and Fig. 1(2), they incorporate a traditional model, but not in Fig. 1(3). Later they show results for a cyclic method that incorporates a physical model, but it's not clear how that was done.
- The authors claim that they are using "a new method for creating testing data". However, testing on extrapolating data is not a new method.

Additional Comments:
- Equation (2): I assume the hat should only be above f, not the entire f(x). If not, I don't see this symbol explained or used anywhere else in the paper.
- h_t in equation (3) and (4) are not explained in the text
- Fig. 1(1) and Fig. 1(3) are not mentioned in the text
- Figure 3 and 4 are hard to understand. Instead of State 0/1/2, it should say angle / angular velocity / acceleration. There should be a table to accompany this figure, so that it's easier to compare the models. As is, it's very hard to compare e.g. Physics-Ensemble-Dense with Dense-Ensemble-RNN.
- There are plenty of typos and a general lack of clarity. E.g. this sentence: "The  alternative  of  performing  end-to-end  of  all  the  models  at  once  is  an  significant  alternative."
- Add reference for original Ljung-Box test


**Experience Assessment:**

I have read many papers in this area.

**Review Assessment: Checking Correctness Of Derivations And Theory:**

I assessed the sensibility of the derivations and theory.

**Review Assessment: Checking Correctness Of Experiments:**

I assessed the sensibility of the experiments.

**Review Assessment: Thoroughness In Paper Reading:**

I read the paper at least twice and used my best judgement in assessing the paper.

---

### Decision · Program_Chairs · 2019-12-19

**Decision:**

Reject

**Comment:**

All reviewers agree that the paper is to be rejected, provided strong claims that were not answered. In this form (especially with such a title) it could not be published (it is more of a technical/engineering interest).